# CRYSTAL: Illuminating LLM Abilities on Language and Code

**Tianhua Tao**[†‡]**, Junbo Li**[†]**, Bowen Tan**[¶]**, Hongyi Wang**[¶]**,**
**William Marshall**[§*]**, Bhargav M. Kanakiya**[§]**, Joel Hestness**[§]**, Natalia Vassilieva**[§]**,**
**Zhiqiang Shen**[†]**, Eric P. Xing**[†¶] **& Zhengzhong Liu**[†]

[†]Mohamed bin Zayed University of Artificial Intelligence
Abu Dhabi, United Arab Emirates
{junbo.li, zhiqiang.shen, eric.xing, hector.liu}@mbzuai.ac.ae

[‡]University of Illinois Urbana-Champaign
Champaign, Illinois, United States
{tianhua3}@illinois.edu

[¶]Carnegie Mellon University
Pittsburgh, Pennsylvania, United States
{btan2, hongyiwa, epxing}@andrew.cmu.edu

[§]Cerebras Systems
Sunnyvale, California, United States
william.fyfe.marshall@gmail.com, {bhargav.kanakiya, joel, natalia}@cerebras.net

## Abstract

Large Language Models (LLMs) specializing in code generation (which are also often referred to as code LLMs), *e.g.*, StarCoder and Code Llama, play increasingly critical roles in various software development scenarios. It is also crucial for code LLMs to possess both code generation and natural language abilities for many specific applications, such as code snippet retrieval using natural language or code explanations. The intricate interaction between acquiring language and coding skills complicates the development of strong code LLMs. Furthermore, there is a lack of thorough prior studies on the LLM pretraining strategy that mixes code and natural language. In this work, we propose a pretraining strategy to enhance the integration of natural language and coding capabilities within a single LLM. Specifically, it includes two phases of training with appropriately adjusted code/language ratios. The resulting model, CRYSTAL, demonstrates remarkable capabilities in both domains. Specifically, it has natural language and coding performance comparable to that of Llama 2 and Code Llama, respectively. CRYSTAL exhibits better data efficiency, using 1.4 trillion tokens compared to the more than 2 trillion tokens used by Llama 2 and Code Llama. We verify our pretraining strategy by analyzing the training process and observe consistent improvements in most benchmarks. We also adopted a typical *application adaptation phase* with a code-centric data mixture, only to find that it did not lead to enhanced performance or training efficiency, underlining the importance of a carefully designed data recipe. To foster research within the community, we commit to open-sourcing every detail of the pretraining[1], including our training datasets[2], code[3], loggings[4] and 136 checkpoints[5] throughout the training.

---

[*]Work done at Cerebras Systems.

[1]Webpage: https://www.llm360.ai/#crystal

[2]Datasets: https://huggingface.co/datasets/LLM360/CrystalCoderDatasets

[3]Code: https://github.com/LLM360/crystalcoder-train

[4]Wandb: https://wandb.ai/llm360/CrystalCoder

[5]Model weights: https://huggingface.co/LLM360/CrystalCoder and https://huggingface.co/LLM360/CrystalChat

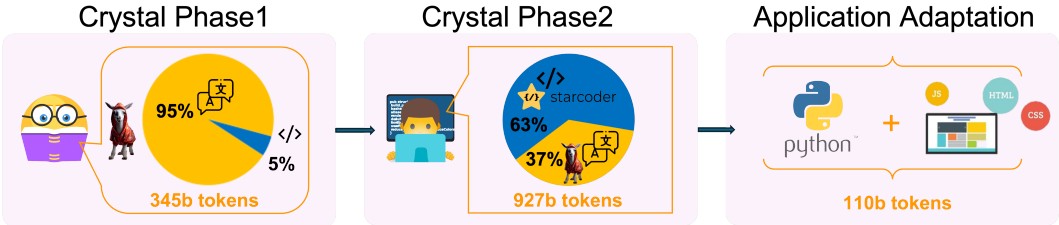

Figure 1: The multi-phase training process for CRYSTAL.

# 1 Introduction

Large Language Models (LLMs) for code generation (*i.e.*, code LLMs), such as Codex (Chen et al., 2021a), StarCoder (Li et al., 2023a), and Code Llama (Roziere et al., 2023), are advancing rapidly due to their strong capability in generating code-related content (*e.g.*, functions), which helps improve the efficiency of software engineers and developers (Cognition Labs, 2024; Chen et al., 2021b; Li et al., 2023a; Roziere et al., 2023). These LLMs excel at generating functions and designing web page components based on engineers' instructions (*e.g.*, *"Return True if all numbers in the list L are below threshold T."*) (Calò & De Russis, 2023). However, the abilities of code-oriented LLMs are constrained in development contexts that necessitate interpreting high-level human instructions (*e.g.*, through prompts or function descriptions) and producing com-

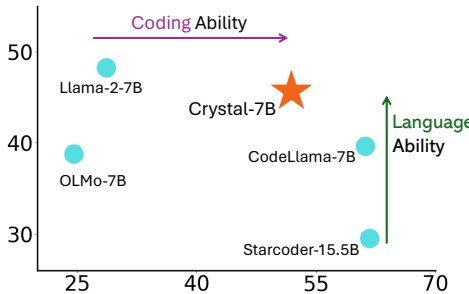

Figure 2: CRYSTAL shows a good balance of language and coding abilities. The *y*-axis is the average over ARC-C, HellaSwag, MMLU, and GSM8K. The *x*-axis is the average of MBPP and HumanEval.

prehensive, structured code accompanied by natural language documentation. Examples of such scenarios include solving GitHub issues (Jimenez et al., 2023), searching for code snippets based on natural language queries, generating entire Python libraries (which include their complete code along with documentation and tutorials (Liu et al., 2023a; Luo et al., 2024)), or developing source code for websites, *e.g.*, "Create a ticketing platform for travelers" (Calò & De Russis, 2023).

This underscores the ambition to create LLMs proficient in both natural language processing and coding. Achieving this goal, however, is non-trivial. For instance, Code Llama, despite being continuously pretrained with code datasets on top of Llama2, suffers from catastrophic forgetting of natural language capabilities. In open-sourced LLMs, we observe a prevalent issue: most models are tailored to specialize in either language or code, not both. For example, StarCoder is exclusively trained on code datasets accompanied by function documentation, thus limiting its exposure to varied natural language data. This trend indicates a notable gap in the design of most open-source LLMs, where there's a lack of a comprehensive curriculum that addresses both coding and natural language processing.

Therefore, we are intrigued by the following research question: "*Can an LLM efficiently obtain both language and coding abilities?*" Existing studies have shown that the simultaneous acquisition of coding and language capabilities by LLMs is governed by complex dynamics: these skills may either conflict (Li et al., 2023a; Roziere et al., 2023) or complement (Ma et al., 2024) each other, influenced by the data recipe and the model's learning phase.

In this work, we propose a pretraining strategy designed specifically for code LLMs. Our strategy is inspired by techniques such as multi-phase pretraining, curriculum learning (Bengio et al., 2009), continuous pretraining (Roziere et al., 2023), and multi-language training, and has two phases. We start the pretraining process with a data mixture of 95% natural language and 5% code. In the second phase, the data mixture is enriched to include 63% code data alongside 37% natural language. This two-phase design mimics the human learning process, where the acquisition of general language knowledge precedes the development

of coding skills, aiming to replicate this learning sequence. Pretraining using our strategy yields CRYSTAL, a code LLM that exhibits strong ability across both natural language (*e.g.*, common sense reasoning) and code generation. Our strategy also demonstrates good data efficiency. That is, CRYSTAL, pretrained with 1.4 trillion tokens, performs comparably to Llama 2 and Code Llama, each pretrained with more than 2 trillion tokens.

Throughout the pretraining process, we continuously tracked the model's performance on downstream benchmarks, observing steady enhancements in both language and coding abilities across the two training phases. Despite a slight performance decline due to the distribution shift between Phase 1 and Phase 2, performance in Phase 2 swiftly recovers and surpasses that of Phase 1. Additionally, implementing an experimental *application adaptation phase*, aimed at further enhancing coding abilities by incorporating an increased percentage of code data, could potentially boost performance. This phase is inspired by the Python-specialized pretraining phase of Code Llama and StarCoder (Roziere et al., 2023; Li et al., 2023a). Contrary to expectations, we observe mixed results from this phase, including a decline in language ability but marginal improvement in coding performance (see § 5.2), underscoring the necessity for a carefully crafted data strategy.

Conducting thorough ablation studies for the entire pretraining process is computationally daunting. To mitigate these challenges, we embrace the principles of the LLM360 initiative (Liu et al., 2023b), ensuring full transparency in our pretraining process to support further scientific exploration and discoveries by the community. We release our training and fine-tuning datasets, source code for training, fine-tuning, and data preprocessing, and 152 intermediate model checkpoints. We also release a chat version, fine-tuned from CRYSTAL, namely CRYSTALCHAT, for user convenience.

## 2 Related Work

**Open-source LLMs.** The prevailing approach to developing modern LLMs involves a two-step process: pretraining followed by fine-tuning. The extensive pretraining stage may involve using synthetic data, as demonstrated by the Phi series models (Gunasekar et al., 2023; Li et al., 2023b). However, the high-quality synthetic datasets used in the Phi models are not publicly available, whereas we aim to make all our training details public and reproducible. Additionally, much of the pretraining is conducted on vast datasets comprising trillions of tokens that encapsulate nearly all available linguistic data. Notable projects in this domain include (Wang & Komatsuzaki, 2021; Andonian et al., 2023; Zhang et al., 2022; Scao et al., 2022; Biderman et al., 2023; Touvron et al., 2023a; Geng & Liu, 2023; Together Computer, 2023a; MosaicML NLP Team, 2023; Almazrouei et al., 2023; Touvron et al., 2023b; Bai et al., 2023; Jiang et al., 2023; 01.ai, 2023; Bi et al., 2024; Groeneveld et al., 2024). Of these, Pythia (Biderman et al., 2023), LLM360/Amber (Liu et al., 2023b) and OLMo (Groeneveld et al., 2024) are particularly aligned with our work, sharing an emphasis on the complete reproducibility of LLMs. While Pythia stands out as a pioneering effort, it does not embody the recent advancements observed in training with trillions of tokens and is not specifically tailored for code. Amber and OLMo, although newer, are also designed as general-purpose models. Our CRYSTAL utilizes advanced strategic data and training strategies to create a strong open-source model that excels in coding while also demonstrating strong overall capabilities.

**Code LLMs.** Applications at the core of fields such as software development engineering (Fan et al., 2023) place a significant demand on language models equipped with specialized code intelligence. Furthermore, models that are extensively trained on code datasets demonstrate enhanced reasoning capabilities and exhibit superior performance in logical tasks, including mathematics (Ma et al., 2024; Fu et al., 2022). Motivated by the needs of both practical applications and research, code-oriented large models are increasingly gaining focus (Chen et al., 2021a; Li et al., 2022; Wang et al., 2023; Luo et al., 2023; Nijkamp et al., 2022; Li et al., 2023a; Roziere et al., 2023; Guo et al., 2024). The roles of code data differ across various works. StarCoder (Li et al., 2023a) exclusively trains on code data, at the expense of general natural language understanding. DeepSeek Coder (Guo et al., 2024) incorporates more natural language into its pretraining, yet remains predominantly focused

on code, with minimal enhancement in natural language benchmarks. Code Llama (Roziere et al., 2023) continues pretraining on code data atop Llama (Touvron et al., 2023a;b), which leads to a forgetting issue that sacrifices general natural language understanding abilities. WizardCoder (Luo et al., 2023) focuses on code instruction finetuning, where the ultimate performance significantly relies on the foundational pretrained model. In contrast, CRYSTAL blends natural language and code data equally in pretraining, aiming to forge a strong code model with enhanced natural language capabilities.

**Multi Stage Training.**  Several LLM training efforts have also documented multi-stage approaches. For example, the XGen project (Nijkamp et al., 2023) includes a relatively short second stage, comprising 4% of the total training tokens, which is similar to our adaptation stage. Our work further demonstrates a successful strategy for mixing two domains during the major training stages. InternLM (InternLM Team, 2023) also reports using a multi-stage training method, though the details of these stages have not been disclosed.

## 3    Model Training

Drawing on the principles of coarse-to-fine methodologies for achieving domain adaptation without catastrophic forgetting, we design two phases in the pretraining process of Crystal. In the first phase, the model is expected to acquire a broad spectrum of general language capabilities. In the second phase, we introduce coding ability into the model, ensuring that this augmentation does not compromise its existing natural language abilities. Table 1 summarizes the configurations of the phases.

**Configuration.**  The architecture of CRYSTAL is adapted from prior work such as GPT-2 (Radford et al., 2019), GPT-NeoX (Andonian et al., 2023), Llama (Touvron et al., 2023a) and BTLM (Dey et al., 2023), featuring decoder-only models comprising 32 layers. The model is trained on a non-GPU hardware architecture, using the Cerebras Condor Galaxy 1 (CG-1) (Cerebras, 2023). Taking advantage of the memory layout, the model can be trained efficiently using LayerNorm without RMSNorm. We incorporate a novel enhancement known as maximal update parameterization ($\mu P$), as described by Yang et al. (2022), deciding layer-wise learning rate, batch size, Adam coefficient, etc. We include all the final hyperparameters adjusted by $\mu P$ in Table 1. The training time is 37 days on 16 CS-2 nodes.

**Phase 1 for Fundamental Language Ability.** In the first phase, we focus on imbuing the model with foundational natural language understanding, utilizing 345B tokens sampled from SlimPajama (Soboleva et al., 2023) dataset. This dataset, primarily composed of natural language texts, includes a modest portion (approximately 5%) of coding data sourced from GitHub, subtly introducing the model to programming concepts. We expect this phase to establish a baseline comprehension of natural language, underpinning the model's subsequent specialization in code. The rationale behind starting with natural language is inspired in curriculum learning principles, positing that mastering the intricacies of natural language is a prerequisite for tackling the structured complexity of programming languages.

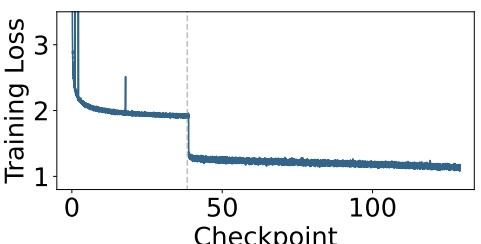

Figure 3: Pretraining loss curve. The gray dashed line divides Phase 1 and 2. We do not observe many major loss spikes; if observed, we recovered by skipping specific data batches.

**Phase 2 for Coding Ability.**  In the second phase, we expand the model's domain by integrating a 63% code data mixture, drawing from a broad spectrum of programming languages from the Stack (Li et al., 2023a) dataset (following the StarCoder mixture), resulting in a total of 927B tokens. This inclusion aims to transit the model from its natural language base towards a more specialized understanding of code syntax and logic. In the meanwhile,

by keeping a significant 37% of general language, we intend to prevent the catastrophic forgetting of general language ability.

|  | Phase 1 | Phase 2 | Phase Adaptation |
|---|---|---|---|
| warm-up steps | 86 | 86 | 276 |
| total steps | 79721 | 214387 | 27590 |
| max LR | 0.012 | 0.0087825 | 0.002 |
| min LR | 0.0086628 | 0.00013679 | 0.0002 |
| optimizer |  | AdamW |  |
| beta1 |  | 0.9 |  |
| beta2 |  | 0.95 |  |
| epsilon |  | 1e-9 |  |
| weight decay |  | 0.1 |  |
| gradient clip |  | 1.0 |  |
| batch size |  | 2112 |  |
| sequence length |  | 2048 |  |
| trained tokens (current phase) | 0.345T | 0.927T | 0.1T |
| trained tokens (accumulated) | 0.345T | 1.272T | 1.372T |

Table 1: Pretraining configuration. We choose the warm-up steps to be approximately 0.1% of the total steps in Phase 1. For Phase 2, we reuse the same numbers. In Adaptation Phase, we set it to be 1% of the total steps.

**Phase Adaptation.** Following prior convention (Li et al., 2023a; Roziere et al., 2023), we conduct additional training to specialize the model on popular language tasks, using a small subset of Python and web-related data (HTML, CSS, and JavaScript) from the Stack dataset, totaling 100B tokens. A small portion (10B tokens) of SlimPajama is added to avoid catastrophic forgetting.

**Finetuning.** We perform tuning on top of Crystal using a collection of open-source datasets, tailored for chat applications, thereby enhancing usability for end users. We denote the resulting model as CRYSTALCHAT. Details can be found in Appendix C.

Figure 3 depicts the training loss curve throughout the training process, highlighting a smooth and stable progression. Loss spikes are observed at the outset of Phase 1. It's important to note that the loss scales vary across the two phases due to the increasing proportion of code data. Code data tokens, which often represent shorter raw untokenized entities such as symbols, digits, and brackets, adhere to specific syntactical rules making them more predictable, resulting in a lower loss-per-token.

# 4 Evaluation

We conduct an extensive evaluation of CRYSTAL across multiple tasks, including language understanding, commonsense reasoning, code generation, and a newly crafted benchmark for website generation. We compare CRYSTAL with models developed around the same time and trained with a comparable number of FLOPs. However, some other open-weight models, such as Mistral (Jiang et al., 2023) and Mixtral (Jiang et al., 2024), do not disclose the size of their training data, making them unsuitable for direct comparison in studying the effect of data curriculum. For similar reasons, we also do not compare CRYSTAL with commercial endpoints.

## 4.1 Evaluating Natural Language Abilities

We evaluate Crystal's language ability on a key set of benchmarks maintained by EleutherAI[6]. We show the benchmark results on a variety of aspects in natural language, including **Reasoning**: Hellaswag (Zellers et al., 2019), ARC (Clark et al., 2018), Winogrande (Sakaguchi et al., 2021), PIQA (Bisk et al., 2020); **Question Answering**: Open-BookQA (Mihaylov et al., 2018), RACE (Lai et al., 2017), BoolQ (Clark et al., 2019),

---

[6]https://github.com/EleutherAI/lm-evaluation-harness

| | CRYSTAL | | | Other Open Source Models | | | |
| | Phase 1 | Phase 2 | Adapt. | Llama 2 | Code Llama | OLMo | StarCoder15.5B |
|---|---|---|---|---|---|---|---|
| **Natural Language Benchmarks** | | | | | | | |
| ARC-easy (0-shot) | 64.73 | 70.75 | 67.34 | **74.50** | 62.29 | 68.51 | 50.17 |
| ARC-challenge (0-shot) | 37.54 | 42.58 | 38.91 | **46.16** | 35.24 | 40.27 | 27.73 |
| ARC-challenge (25-shot) | 42.83 | 47.44 | 47.01 | **53.33** | 42.75 | 45.93 | 32.16 |
| Openbook QA (0-shot) | 39.60 | 41.20 | 39.80 | **44.20** | 36.80 | 42.60 | 32.20 |
| TruthfulQA (5-shot) | 38.96 | 36.47 | 35.91 | 38.95 | 37.19 | 35.92 | **41.36** |
| MMLU (0-shot) | 28.05 | 42.46 | 42.33 | 41.71 | 34.76 | 28.19 | 27.55 |
| MMLU (5-shot) | 25.72 | 48.42 | 48.78 | 46.40 | 39.98 | 28.12 | 28.45 |
| HellaSwag (0-shot) | 69.65 | 72.89 | 70.35 | **75.92** | 62.80 | 75.56 | 46.65 |
| HellaSwag (10-shot) | 71.62 | 74.38 | 71.97 | **78.5** | 64.74 | 77.12 | 48.36 |
| RACE (0-shot) | 38.57 | 38.18 | 38.18 | **39.52** | **39.52** | 38.37 | 31.67 |
| PIQA (0-shot) | 75.84 | 78.07 | 76.77 | 78.78 | 72.58 | **79.92** | 65.61 |
| COPA (0-shot) | 86.00 | 83.00 | 80.00 | 87.00 | 80.00 | **88.00** | 67.00 |
| BoolQ (0-shot) | 66.64 | 74.43 | 72.36 | **78.07** | 74.65 | 72.66 | 57.16 |
| Winogrande (0-shot) | 63.14 | 67.01 | 65.51 | **69.38** | 65.51 | 67.24 | 55.10 |
| Winogrande (5-shot) | 64.80 | 68.82 | 67.40 | **73.64** | 65.75 | 68.90 | 56.04 |
| GSM8K (5-shot) | 2.12 | 12.36 | 10.39 | **14.71** | 11.15 | 4.09 | 9.02 |
| **Code Benchmarks** | | | | | | | |
| HumanEval (pass@1) | 7.44 | 23.90 | 28.38 | 13.05 | 30.06 | 14.02 | **33.63** |
| HumanEval (pass@10) | 14.64 | 45.12 | 52.76 | 22.61 | 58.36 | 24.56 | **59.38** |
| MBPP (pass@1) | 8.92 | 30.99 | 36.37 | 20.09 | 39.20 | 14.40 | **52.7** |
| MBPP (pass@10) | 17.24 | 58.62 | 56.37 | 34.69 | 64.00 | 26.42 | **65.44** |
| Multipl-e Bash (pass@1) | 0 | **10.76** | 6.96 | 2.53 | 10.13 | 1.26 | 10.12 |
| Multipl-e C++ (pass@1) | 6.83 | 24.22 | 23.60 | 6.83 | 26.08 | 13.04 | **29.81** |
| Multipl-e C# (pass@1) | 3.17 | 17.09 | 17.09 | 6.32 | **23.41** | 8.86 | 20.88 |
| Multipl-e Java (pass@1) | 3.17 | 22.79 | 27.21 | 11.39 | **33.54** | 13.29 | 29.74 |
| Multipl-e JS (pass@1) | 9.94 | 29.19 | 29.81 | 12.42 | **35.40** | 14.91 | 31.05 |
| Multipl-e PHP (pass@1) | 4.97 | 20.497 | 20.50 | 9.94 | 24.22 | 7.45 | **27.32** |
| Multipl-e TS (pass@1) | 10.06 | 25.15 | 30.18 | 13.21 | 32.70 | 12.57 | **33.96** |

Table 2: Natural language and code generation evaluation results. The best scores for CRYSTAL are underlined. The best scores across all compared models are **bold**. All models in the table are around 7B in size, except that StarCoder is a 15.5B model.

COPA (Reddy et al., 2019); **General Knowledge**: MMLU (Hendrycks et al., 2021b); **Basic Arithmetic**: GSM8K (Cobbe et al., 2021); **Truthfulness**: TruthfulQA (Lin et al., 2021).

The results of these evaluations are detailed in Table 2. More results are available in our Weights & Biases public dashboard online[7]. Our results demonstrate that CRYSTAL achieves competitive performance across a range of language tasks, with Phase 2 checkpoints outperforming other popular open-source LLMs, *e.g.*, Llama 2, Code Llama, OLMo, and StarCoder-15.5B on several benchmarks even with fewer number of training tokens (Touvron et al., 2023b; Roziere et al., 2023; Groeneveld et al., 2024; Li et al., 2023a).

### 4.2 Evaluating Code Generation

For code generation, we evaluate the models on three benchmarks, i.e., HumanEval (Chen et al., 2021b), MBPP (Austin et al., 2021) and Multipl-e (Cassano et al., 2022), where HumanEval and MBPP are measuring functional correctness for synthesizing programs from docstrings, consists of more than 1,000 Python programming problems in total. Multiple-e is a translation of HumanEval from Python to 18 programming languages. We omitted APPS (Hendrycks et al., 2021a) and DS-1000 (Lai et al., 2022) from our evaluation. The exclusion was due to the significant discrepancies in baseline model scores produced by our setup compared to those reported in their respective papers, primarily attributed to differences in prompts and evaluation configurations. The results can be found in Table 2.

**WebMC: A New Benchmark on Web Programs.** In addition to the three benchmarks above, we introduce a new benchmark, *WebMC*, created for evaluating coding abilities related to website development. This dataset was generated using GPT-4 and manually curated into 600 multiple-choice questions regarding website understanding, edit-

---

[7] https://wandb.ai/llm360/CrystalCoder?nw=hdze3lfpuer

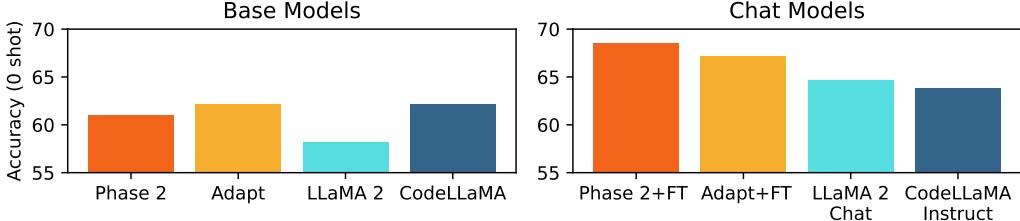

Figure 4: Evaluation results comparison across different models for zero-shot WebMC

ing and generation. WebMC has 3 types of tasks, including: (1) Generation: generating a website according to specifications (e.g., with a map widget), (2) Editing: changing the font-family/font-color, and (3) Understanding: Answer questions about element details, such as the content in the navigation bar. The dataset is available at https://huggingface.co/datasets/LLM360/WebMC.

We conduct an evaluation of CRYSTAL, CRYSTALCHAT against Llama 2, Code Llama, and their instruction-tuned variants. As shown in Figure 4, CRYSTAL demonstrates superior average performance in this comparison.

# 5 Analysis and Discussion

The evaluation results show that with an appropriate data curriculum, it's possible to train a model to excel in both language and code, sometimes even outperforming models with larger token budgets on certain benchmarks. In this section, we delve into an analysis of intriguing aspects observed during training.

---

**GSM8K Example**

**Instruction:** Question: James decides to run 3 sprints 3 times a week. He runs 60 meters each sprint. How many total meters does he run a week?
**Ground Truth:** He sprints 3*3=9 times, so he runs 9*60=540 meters #### 540

**Phase 2 Response:** 180

**Phase 2+FT Response:** James runs 3 sprints * 60 meters each sprint = 180 meters per sprint. He runs this for 3 times a week, so he runs 180 meters per sprint * 3 times a week = 540 meters a week. The answer is 540.

**MBPP Example**

**Instruction:** Write a function to remove first and last occurrence of a given character from the string.

**Phase 2 Response:**
```
def remove_Occ(string, char):
    # TODO: your code here
```

**Phase 2+FT Response:**
```
def remove_Occ(s, c):
    return s.replace(c, '', 1) + s.replace(c, '', -1)
```

---

Table 3: Examples of generation tasks enhanced by finetuning

## 5.1 Effect of Language and Code Mixing in Pretraining

In Fig. 5, we observe that during phase transition, there is a notable drop in performance due to the abrupt change of domain on most benchmarks[8] (ARC-C, Winogrande, and more in Appendix D.2). Within each phases, most benchmark scores for all intermediate checkpoints are rising smoothly, surpassing Phase 1 performance at the end of Phase 2.
For instance, in Hellaswag 10-shot benchmark, a decrease in score from approximately 71 to 67 after the initial 1k steps into Phase 2. It requires about 70% of Phase 2 to restore

---

[8]Except for MMLU, which fluctuates randomly during Phase 1.

|  | Phase 2 | Phase 2+FT | Adapt | Adapt+FT |
|---|---|---|---|---|
| **Natural Language Benchmarks** | | | | |
| ARC-challenge (25-shot) | 47.44 | 51.71 | 47.01 | 50.09 |
| Openbook QA (0-shot) | 41.20 | 42.00 | 39.80 | 38.8 |
| TruthfulQA (5-shot) | 36.47 | 47.29 | 35.91 | 45.13 |
| MMLU (5-shot) | 48.42 | 53.22 | 48.78 | 52.77 |
| HellaSwag (10-shot) | 74.38 | 76.12 | 71.97 | 72.76 |
| Winogrande (5-shot) | 68.82 | 70.64 | 67.40 | 68.19 |
| GSM8K (5-shot) | 12.36 | 28.05 | 10.39 | 27.98 |
| **Code Benchmarks** | | | | |
| HumanEval (pass@1) | 23.90 | 34.12 | 28.38 | 33.29 |
| HumanEval (pass@10) | 45.12 | 65.76 | 52.76 | 69.20 |
| MBPP (pass@1) | 30.99 | 39.40 | 36.38 | 40.20 |
| MBPP (pass@10) | 58.62 | 59.90 | 56.37 | 61.16 |
| Multipl-e Bash (pass@1) | 10.76 | 12.65 | 6.96 | 12.65 |
| Multipl-e C++ (pass@1) | 24.22 | 32.91 | 23.60 | 33.54 |
| Multipl-e C# (pass@1) | 17.09 | 24.05 | 17.09 | 23.41 |
| Multipl-e Java (pass@1) | 22.79 | 34.81 | 27.22 | 36.70 |
| Multipl-e JS (pass@1) | 29.19 | 31.67 | 29.81 | 42.23 |
| Multipl-e TS (pass@1) | 25.16 | 34.59 | 30.19 | 33.33 |

Table 4: Evaluation results for finetuned (FT) models based on Phase 2 and Adaptation Phase (Adapt). More results on finetuned and instruction-following open source models can be found in Table 7.

performance levels, but after that the model can surpass the previous best value. The significant performance dips in these tasks at the beginning of Phase 2 indicate that the abrupt shift in data distribution may be harmful. In reflection, a more gradual transition to more code data may foster smoother adaptation.

For MMLU, the score is almost around a random baseline at 25% in Phase 1 (with 5% code data). A faster growing trend is observed in Phase 2 (with 63% code data). The final MMLU score after two phases is even higher than strong baselines such as Llama 2. There seems to be an interesting interaction between language and code causing this phenomenon. One potential explanation can be data contamination. However, our pretraining datasets are SlimPajama (a deduplicated version of RedPajama) and StarCoder data (a dataset filtered from Github). We argue that the same or similar datasets have been used by other open-source models of comparable scales(Touvron et al., 2023a; Li et al., 2023a; Groeneveld et al., 2024), yet similar performance gain has not been observed in these models. This disparity in performance may suggest that our model's success cannot be fully attributable to data content. We hypothesize that it can be attributed to the unique combination of data selection, model architecture, and training curriculum. Although a full pretraining ablation study is costly, our approach has been independently verified by the Snowflake Arctic team, which found that a similar three-stage curriculum learning strategy can improve key evaluation metrics like MMLU (Snowflake AI Research, 2024). We also hope that our intermediate checkpoints will enable further study in this area.

We observe that supervised fine-tuning will boost the benchmark score significantly for generation tasks, such as MBPP and GSM8K. By inspecting the output (Table 3), we notice that the model tend to generate complete output, not mimicking the input question and follow coding instructions closely. Furthermore, the fine-tuned model will sometimes conduct chain-of-thought style generation, which is very helpful at solving arithmetic questions. Though we do not fully rule out the chances of data contamination.

Overall, the effect of mixing surpass our expectation. The performance on language and code are both high and achieve a good balance, as depicted in Figure 2.

## 5.2  Adaptation Phase: Cost and Gain

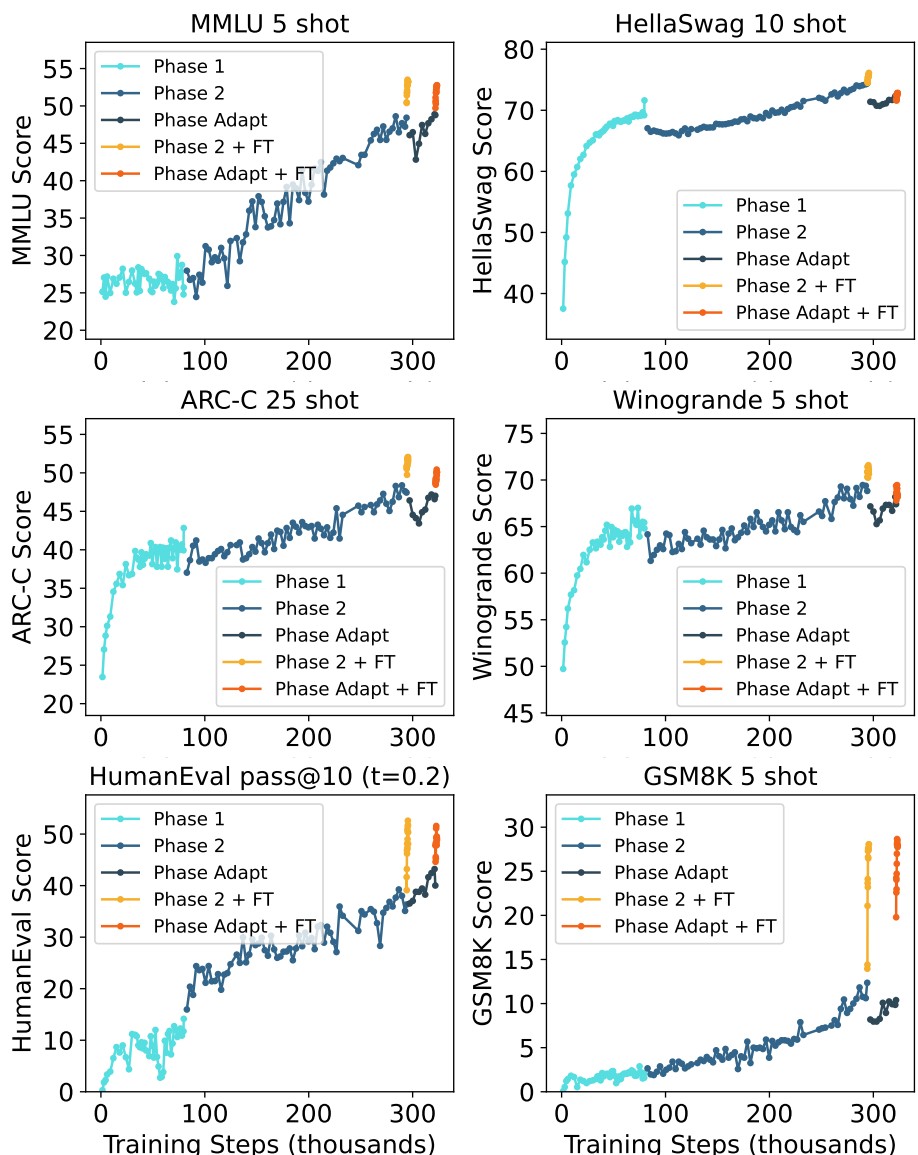

Figure 5: Benchmark scores for all intermediate checkpoints across phases. Contrary to prior work, our Adaptation Phase does not improve the model. Instruction finetuning generally boost the model performance as expected.

Tasks that are more directly connected to the adaptation dataset (Python/JS/HTML/CSS), such as HumanEval/MBPP (Python-related) and Multipl-e JS/TS (JavaScript-related), show expected enhancements. For programming languages that are not reinforced, we observe mixed outcomes: some abilities improve (*e.g.*, Java), while others suffer from certain declines (*e.g.*, Bash and C++). However, for the majority of natural language tasks, the Adaptation Phase causes a significant performance decrease[9]. We hypothesize that the sudden decrease in the data ratio for language data may be responsible for this degradation

At a first glance, it seems that the Adaptation Phase brings expected specialization. However, if we take into account of further finetuning, the adaptation gains are even out afterwards. Table 7 shows that, the model after finetuning Phase 2 (Phase 2+FT) and Phase Adaptation

---

[9]Though the MMLU score remains unaffected.

(Phase Adapt+FT), ends up with similar performance. Contrary to the findings in prior work (Li et al., 2023a; Roziere et al., 2023), we observe that the marginal benefits brought by this phase is not worth the increased training cost, at least with this data curriculum.

## 5.3 Training Efficiency

Given that increased compute budget typically leads to better outcomes, we argue that LLMs should be evaluated based on compute efficiency in additional to final performance. Viewing the results of CRYSTAL through the lens of training efficiency can further validate the effectiveness of the training curriculum.

In Figure 6 we see the trade-off between total FLOPs[10] spent on language data during pretraining and downstream performance. We see that CRYSTAL phases 1 and 2 models achieve stronger performance efficiently, even when compared to very strong baselines such as Llama 2.

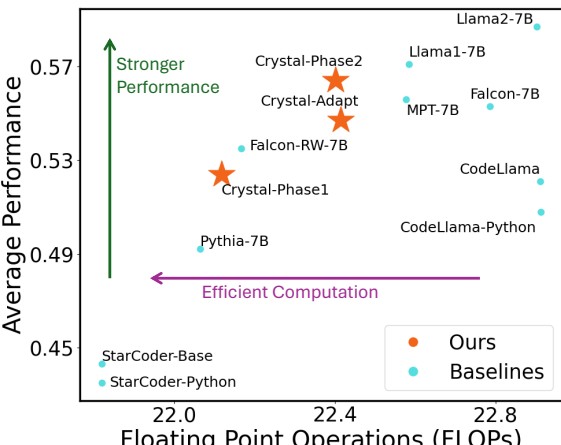

Figure 6: The trade-off between the FLOPs (in log scale) required for language training and the average performance of ARC, MMLU, and TruthfulQA.

## 6 Conclusion

In this work, we present a multi-phase LLM pretraining method designed to encode natural language and coding abilities into a single model. Through this approach, we obtain a model CRYSTAL, achieving natural language and coding performance on par with Llama 2 and Code Llama, respectively. By tracking, observing and analyzing model performance throughout the pretraining, as well as a study of an additional adaptation phase, we obtain and present insights of the interplay of language and coding ability acquisition during the model training, highlighting the importance of data curriculum design.

Though we have included careful analysis on the training process, it remains challenging to verify each design choice and explain every observed phenomenon during pretraining, largely due to constraints in computational resources. We will address the following limitation and release all our training artifacts, inviting the community to collaborate in overcoming these challenges:

**Necessity of a Smooth Transition Between Phases.** Our examination in Section 5.1 reveals a slight performance drop when transitioning from Phase 1 to Phase 2 in certain benchmarks. This observation hints at the potential importance of ensuring a smooth transition between training phases. Further investigation and validation of this hypothesis could further enhance our training methodology.

**Impact of Code on Enhancing Reasoning Abilities.** Notably, our second phase, which incorporates 67% code data, yielded unexpectedly high scores on the MMLU, a benchmark for natural language. This outcome suggests that structured code data may also boost language capabilities. A more definitive confirmation of this hypothesis could be obtained by re-running the second phase while omitting all code data, allowing for a direct assessment of its impact on learning outcomes.

## Acknowledgement

We thank the anonymous reviewers for their insightful comments and suggestions.

---

[10]Estimated by FLOPs per token times number of tokens of language data.

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

## A  Responsible Research

LLM360 is created with the mission to train and release open-source large language models to foster transparency, trust, and collaborative research. While large language models have demonstrated promise in advancing numerous domains throughout commercial and academic settings, the technology is still relatively poorly understood. Due to the significant capital requirements to training and experimentation with LLMs, many learnings in the space happen behind closed doors. The lack of knowledge transfer will have negative effects for the ecosystem as advances will be limited to small groups. To fully realize the potential large language models can deliver, we believe that the core tenets of transparency, trust, and collaboration are paramount to the long term success of the field.

For each model released under LLM360, we will release the datasets, data preparation scripts, training code, numerous intermediate checkpoints, and complete analysis. We prioritize publicly available datasets such as The RedPajama (Together Computer, 2023b) and Refined Web (Penedo et al., 2023) and existing architectures and conventions such as LLaMA (Touvron et al., 2023a) to make our resource relevant and easy to access. By providing the listed artifacts, we hope to promote the reproducibility for all our work to encourage additional research.

Datasets are expensive to curate and are a major competitive advantage for training performant models. By making all data available, our models are fully auditable. We provide clarity on all pretraining sources, the ethical manner in which data was sourced, and the actual data. Releasing checkpoints from the entire training process enables fine grained research into training dynamics (Qian et al., 2024) which would otherwise be restricted to those with the financial resources to pretrain models. We believe that the future should only be constrained by our creativity, not man-made hurdles, and hope that access to our artifacts motivates others to pursue their own creative research unhindered.

**Ethical Use.** LLM360 models openly release our scores on safety evaluations such as Toxigen and TruthfulQA. These scores educate users to the potential risks that using our models may introduce when generating text. We gather our data from reputable sources and apply rigorous filtering to remove harmful data, but we cannot guarantee the outputs of our models will be completely safe. All users should conduct their own testing before adopting our models.

LLM360 models are also trained with coding abilities. When using code generated from large language models, users should always review the output before submitting it into their codebase. Generated code may introduce issues such as insecure code which cannot be eliminated from the model. Users should perform their own safety testing and code reviews before deploying applications.

## B  Model Architecture

The Crystal language models employ a GPT-like architecture, featuring decoder-only models comprising 32 layers. We incorporate a novel enhancement known as maximal update parameterization (muP), as described by Yang et al. (2022), enabling uniformity in hyperparameters including optimization-related hyper-parameters, *i.e.*, *learning rate, batch size, Adam coefficient*, etc., and initialization-related hyper-parameters across models of varying widths. This uniformity facilitates the optimization of hyper-parameters by tuning a smaller, shallow model and directly transferring the optimized settings to the original wider model. Intuitively, muP achieves this by regularizing each linear layer relative to its width, rendering updates "independent" of width.

1. Input embeddings are scaled by `mup_embeddings_scale`.
2. Output logits are scaled by `mup_output_alpha` $\times$ `mup_width_scale`.
3. Attention weights scaling is refined to division by the hidden dimension size $\left( \frac{QK^T}{d} \right)$ instead of its square root $\left( \frac{QK^T}{\sqrt{d}} \right)$. We find this works better under the muP setting in early experiments.
4. Learning rates and weight decay are optimized for different parameter groups:
   - Embedding layer: LR=BASE_LR, WD=BASE_WD.
   - Normalization layers: LR=BASE_LR, WD=0.
   - Other Parameters: LR=BASE_LR $\times$ `mup_width_scale`, WD=BASE_WD.
5. Initialization ranges are determined based on muP hyperparameters.

The muP hyperparameters are set as follows:

- `mup_initialization_standard_deviation`: 0.073
- `mup_embeddings_scale`: 14.6
- `mup_output_alpha`: 2.22
- `mup_width_scale`: 0.0625
- `mup_base_width`: 256

For other architecture choices:

- We use `LayerNorm` instead of `RMSNorm`.
- Rotary position embeddings applied to only the first 25% of hidden dimensions (Black et al., 2022), leaving the other 75% dimensions unchanged.
- Training sequence length is 2048.
- Embedding dimension is 32032.

Full architecture details and comparisons with other models are available in table 5.

## C  Finetuning Details

### C.1  Prompt Format

We introduced four special tokens to the tokenizer and model architecture to enhance instruction handling:

- `<|sys_start|>` — Marks the beginning of a system prompt.
- `<|sys_end|>` — Marks the end of a system prompt.
- `<|im_start|>` — Marks the start of an instruction message.

| | Crystal | Llama 2 |
|---|---|---|
| Layers | 32 | 32 |
| Hidden Dimension | 4096 | 4096 |
| Embedding Dimension | 32032 | 32000 |
| Positional Embedding | Rotary | Rotary |
| Rotary Percentage | 25% | 100% |
| Layer Normalization | LayerNorm | RMSNorm |
| Num Heads | 32 | 32 |
| Activation | SwiGLU | SwiGLU |
| Sequence Length | 2048 | 4096 |
| Batch size | 2112 | 1024 |
| Bias | Linear & LayerNorm | None |
| muP | Yes | No |
| QK Dot Product Scaling | $\frac{QK^T}{d}$ | $\frac{QK^T}{\sqrt{d}}$ |

Table 5: Architecture comparison.

- `<|im_end|>` — Marks the end of an instruction message.

These tokens were integrated into the existing 32032-token vocabulary without necessitating an expansion, leveraging reserved vocabulary slots. The conversation is wrapped by the tokens `` and ``, framing the structure as follows:

```
 <|sys_start|> system prompt <|sys_end|> <|im_start|> first user utterance
  ↪ <|im_end|> first model response <|im_start|> next user utterance <|
  ↪ im_end|> next model response 
```

## C.2 Finetuning Datasets

Table 6 summarizes the datasets we use for finetuning.

| Subset | #Tokens | Avg. #Q | Avg. Q Len | Avg. #R | Avg. R Len |
|---|---|---|---|---|---|
| OASST1-guanaco | 4,464,640 | 1.36 | 38.28 | 1.36 | 271.69 |
| SlimOrca | 225,628,160 | 1.00 | 259.16 | 1.00 | 151.12 |
| ShareGPT | 112,914,432 | 3.28 | 94.53 | 3.64 | 365.81 |
| Evol-ShareGPT | 85,954,560 | 1.00 | 145.99 | 1.00 | 425.17 |
| ChatLogs | 29,337,600 | 3.39 | 95.58 | 3.24 | 191.42 |
| CodeAlpaca | 2,623,488 | 1.00 | 32.46 | 1.00 | 67.68 |
| Rosetta Code | 7,987,200 | 1.00 | 450.09 | 1.00 | 533.52 |
| Evol-CodeAlpaca 1 | 73,803,776 | 1.00 | 210.33 | 1.00 | 437.92 |
| Evol-CodeAlpaca 2 | 34,910,208 | 1.00 | 114.99 | 1.00 | 300.29 |
| WebAlpaca | 43,673,600 | 1.00 | 96.29 | 1.00 | 746.52 |
| General Textbooks | 85,590,016 | Not instruction data | | | |
| Programming Books | 395,628,544 | Not instruction data | | | |
| Total | 1,102,516,224 | | | | |

Table 6: Dataset Statistics. Q stands for Query. R Stands for reply. The summarizes the average number and length of the queries and replies for the datasets. We also included textbook style datasets in the final finetuning dataset.

# D   Evaluation Details

## D.1   Full Evaluation Details for Chat Models

We present the full evaluation results of our models and other open source models in Table 7.

| | CRYSTAL | | | | Other Open Source Models | | |
| --- | --- | --- | --- | --- | --- | --- | --- |
| | Phase 2 | Phase 2+FT | Adapt | Adapt+FT | Llama 2 Chat | CodeLlama-Instruct | OLMo-Instruct |
| **Natural Language** | | | | | | | |
| ARC-easy (0-shot) | 70.75 | 70.33 | 67.34 | 68.65 | 69.61 | 63.09 | 64.14 |
| ARC-challenge (0-shot) | 42.58 | 44.63 | 38.91 | 43.09 | 44.54 | 36.60 | 43.08 |
| ARC-challenge (25-shot) | 47.44 | 51.71 | 47.01 | 50.09 | 53.07 | 43.35 | 47.95 |
| Openbook QA (0-shot) | 41.20 | 42.00 | 39.80 | 38.8 | 43.60 | 37.20 | 45.20 |
| TruthfulQA (5-shot) | 36.47 | 47.29 | 35.91 | 45.13 | 45.30 | 39.23 | 45.52 |
| MMLU (0-shot) | 42.46 | 52.79 | 42.33 | 51.01 | 47.17 | 40.67 | 47.48 |
| MMLU (5-shot) | 48.42 | 53.22 | 48.78 | 52.77 | 48.42 | 42.75 | 48.57 |
| HellaSwag (0-shot) | 72.89 | 73.31 | 70.35 | 70.94 | 75.46 | 63.83 | 78.47 |
| HellaSwag (10-shot) | 74.38 | 76.12 | 71.97 | 72.76 | 78.39 | 66.14 | 79.56 |
| RACE (0-shot) | 38.18 | 41.15 | 38.18 | 41.44 | 43.63 | 40.00 | 40.57 |
| PIQA (0-shot) | 78.07 | 77.86 | 76.77 | 76.17 | 77.26 | 73.93 | 76.71 |
| COPA (0-shot) | 83.00 | 85.00 | 80.00 | 88.00 | 89.00 | 81.00 | 87.00 |
| BoolQ (0-shot) | 74.43 | 82.78 | 72.36 | 80.46 | 79.76 | 73.79 | 78.56 |
| Winogrande (0-shot) | 67.01 | 68.11 | 65.51 | 68.19 | 66.29 | 64.48 | 66.77 |
| Winogrande (5-shot) | 68.82 | 70.64 | 67.40 | 68.19 | 73.08 | 64.32 | 70.24 |
| GSM8K (5-shot) | 12.36 | 28.05 | 10.39 | 27.98 | 18.88 | 15.92 | 11.59 |
| **Code Generation** | | | | | | | |
| HumanEval (pass@1) | 23.90 | 34.12 | 28.38 | 33.29 | 13.26 | 33.53 | 18.90 |
| HumanEval (pass@10) | 45.12 | 65.76 | 52.76 | 69.20 | 24.89 | 59.6[†] | 28.39 |
| MBPP (pass@1) | 30.99 | 39.40 | 36.38 | 40.20 | 17.42 | 38.91 | 15.40 |
| MBPP (pass@10) | 58.62 | 59.90 | 56.37 | 61.16 | 32.12 | 66.7[†] | 27.32 |
| Multipl-e Bash (pass@1) | 10.76 | 12.65 | 6.96 | 12.65 | 1.27 | 10.76 | 0.63 |
| Multipl-e C++ (pass@1) | 24.22 | 32.91 | 23.60 | 33.54 | 9.32 | 31.05 | 13.66 |
| Multipl-e C# (pass@1) | 17.09 | 24.05 | 17.09 | 23.41 | 9.49 | 24.05 | 7.59 |
| Multipl-e Java (pass@1) | 22.79 | 34.81 | 27.22 | 36.70 | 8.23 | 30.38 | 8.22 |
| Multipl-e JS (pass@1) | 29.19 | 31.67 | 29.81 | 42.23 | 14.91 | 31.05 | 1180 |
| Multipl-e PHP (pass@1) | 20.50 | 33.54 | 20.50 | 31.05 | 10.56 | 27.33 | 9.32 |
| Multipl-e TS (pass@1) | 25.16 | 34.59 | 30.19 | 33.33 | 11.95 | 32.70 | 13.21 |

Table 7: Natural language and code generation evaluation results for chat models. The best scores for CRYSTAL are underlined. Numbers with [†] are adopted from original papers. Comparing the results for "Phase 2+FT" and "Adapt+FT", the benefits brought by the Adaptation Phase become insignificant after the finetuning, .

## D.2   Evaluation Results on Intermediate Checkpoints

We present evaluation results on all intermediate checkpoints.

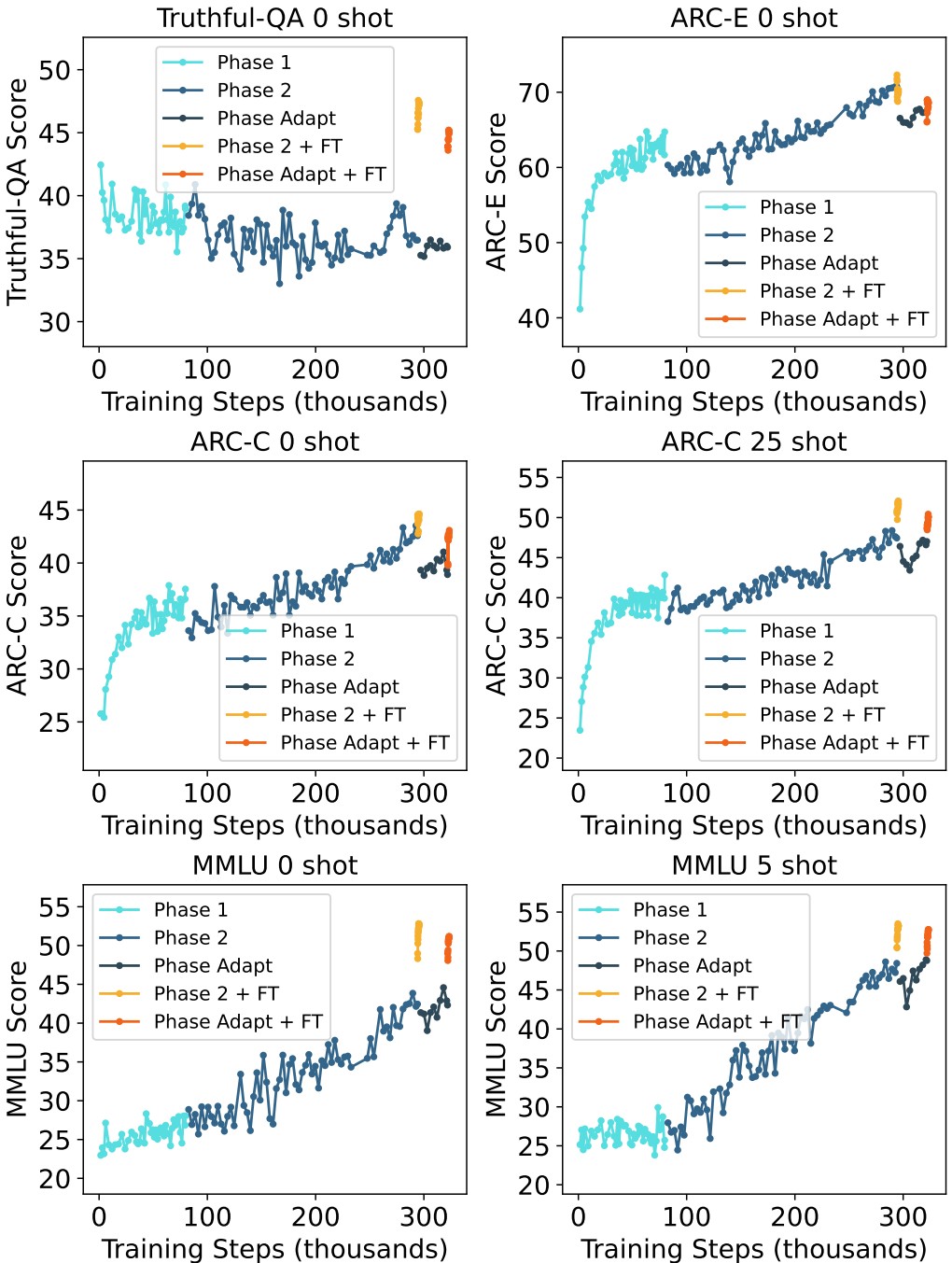

Figure 7: Evaluation results (part 1) on intermediate checkpoints.

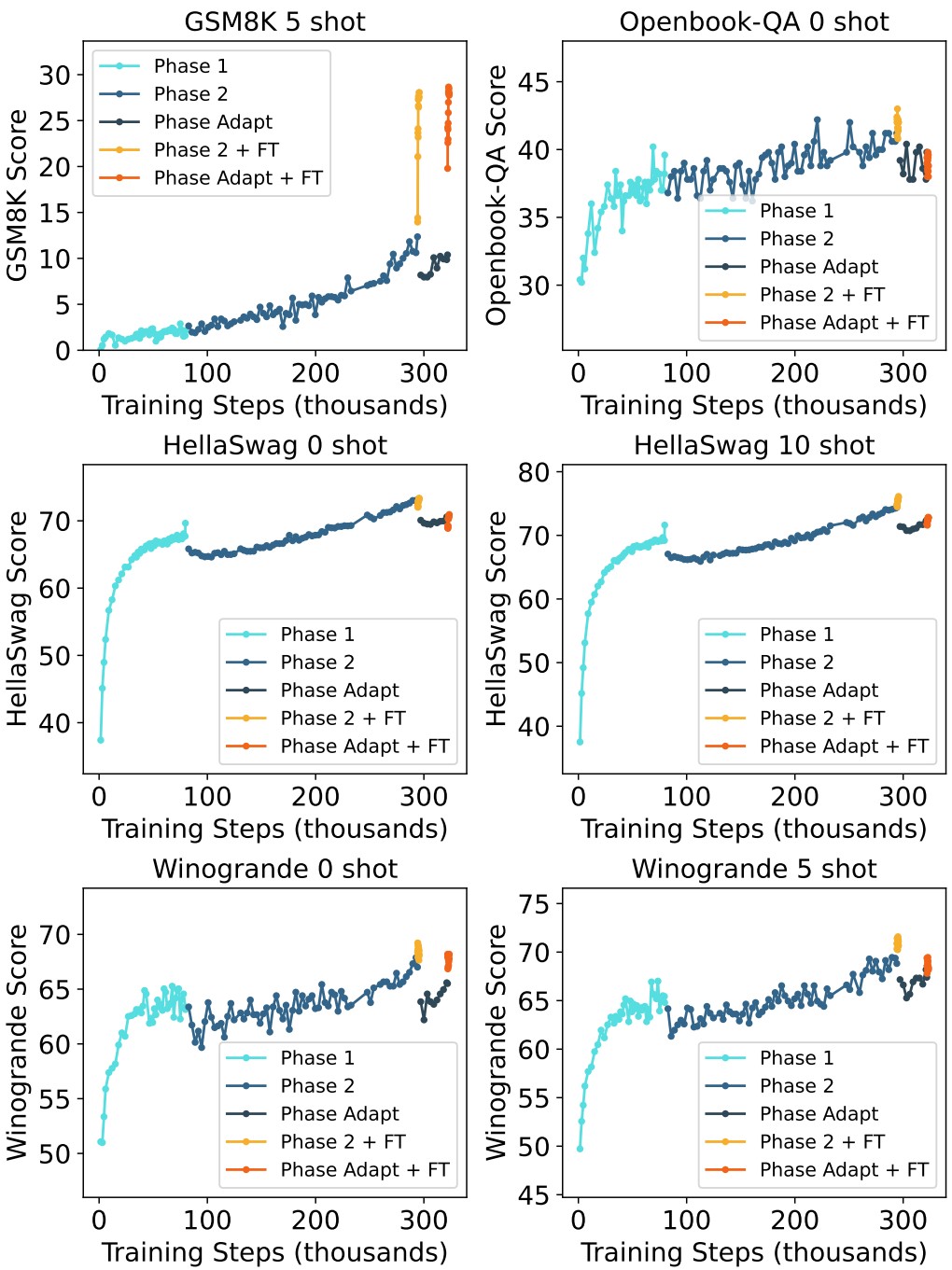

Figure 8: Evaluation results (part 2) on intermediate checkpoints.

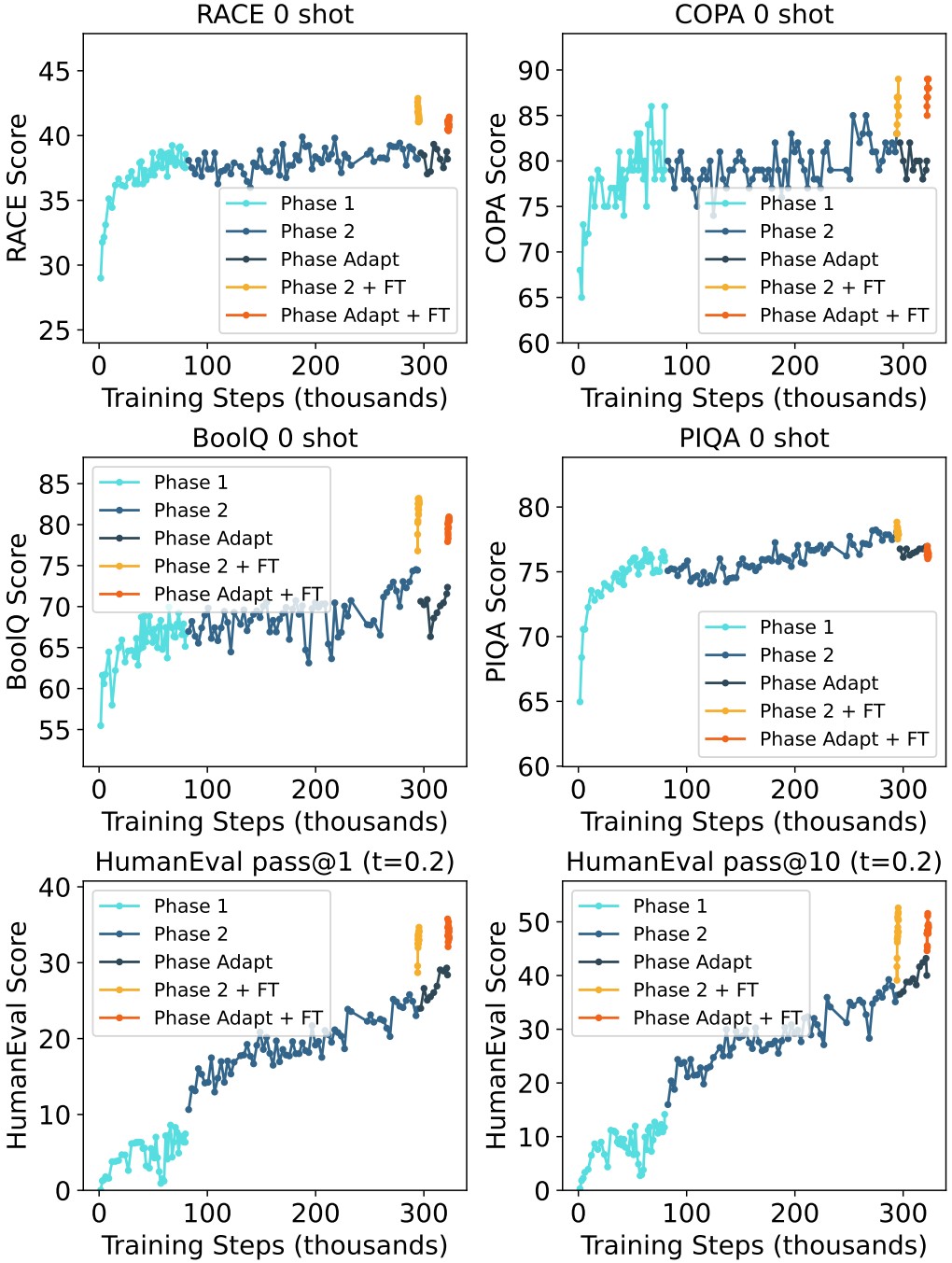

Figure 9: Evaluation results (part 3) on intermediate checkpoints.

