# OpenReview forum: "Crystal: Illuminating LLM Abilities on Language and Code"
_colmweb.org/COLM/2024/Conference — COLM_

### Official Review · Reviewer_zzp6 · 2024-04-22

**Rating:** 7
**Confidence:** 4
**Ethics Flag:** 1

**Summary:**

This paper investigates the integration of natural language and coding capabilities within code LLMs. The authors introduce a novel pretraining strategy that involves two training phases with a carefully adjusted mix of code and natural language data. Their proposed model, Crystal, demonstrates comparable performance in natural language processing and coding tasks to established models like Llama2 and Code Llama, but with greater data efficiency, using 1.4T tokens versus over 2T used by the other models. The effectiveness of the new pretraining approach is supported by detailed analysis of the training process and benchmarking, which show consistent improvements.

**Questions To Authors:**

See weakness 1. Is it possible to see if performance improvements are due to the two-phase training method or if similar outcomes could be achieved with a single-stage training using a combined dataset?

**Reasons To Accept:**

1. The motivation behind this paper is strong, highlighting a critical gap in open-source LLMs which typically excel either in coding or NLP tasks, but not both. This contrasts sharply with more versatile closed-source models like GPT-4.

2. The paper provides detailed insights into model performance through the inclusion of evaluation results from different training checkpoints (Figure 5) and ablation studies (Table 3), clearly illustrating the impact of various training phases on Crystal’s effectiveness across tasks.

3. Comparative analysis reveals that Crystal achieves strong performance on both coding and NL tasks, rivaling the capabilities of similarly sized models in two domain at the same time.

4. The evaluation is comprehensive, particularly in the NLP domain, where Crystal is benchmarked against strong baselines such as LLaMa-2, Code LLaMa, and StarCoder across 11 distinct tasks.

5. Crystal shows notable computational efficiency, achieving strong performance while requiring significantly fewer training tokens than comparable models.

Note: While the paper mentions a commitment to releasing training code and all intermediate checkpoints, which could be a significant contribution, this is not yet verifiable in the review phase without supplementary material. Thus, it is not included as a strength here.

**Reasons To Reject:**

1. It remains unclear whether the observed performance improvements are due to the two-phase training method or if similar outcomes could be achieved with a single-stage training using a combined dataset. The current ablation study does not address this question. Although retraining the full model is resource-intensive, conducting a smaller-scale experiment, perhaps with a 1B parameter model, could validate the effectiveness of the two-phase approach, which is central to this paper’s contributions.

2. While the paper benchmarks Crystal against notable models, it omits comparisons with key baselines like GPT-3.5 and GPT-4, which are recognized for their proficiency in both coding and NL tasks. Additionally, including comparisons with models like Mixtral-8x7B, which also excel in both domains, would enhance the robustness of the evaluation.

3. *[Minor]* The pretraining strategy proposed in this paper, which consists of two distinct phases using different mixtures of NL and code datasets, is too straightforward and lacks significant novelty.

4. *[Minor]* The evaluation methodology appears somewhat biased towards NLP, as the evaluations on the coding aspects (3 tasks) are not as comprehensive compared with on the NLP side (11 tasks). Expanding the evaluation to include additional coding benchmarks such as APPS or DS-1000 could provide a more balanced view of the model's capabilities.

---

> ### Author Rebuttal · Authors · 2024-05-31
>
> Thank you for the thoughtful review and helpful feedback! We respond to individual comments below.
>
> **Ablation Study**\
> We have started a 7B ablation but haven’t finished it due to resource constraints. We discussed the idea of smaller models but:
> - Scaling behaviors for multi-phase training is more complex.
> - MMLU is a key gain in our 7B setup. Yet our 1B models only have MMLU accuracy at random, the general trend is that knowledge of extensive tasks relies strongly on model sizes, hence ablation this way didn’t provide additional insights.
>
> While not a direct comparison, we collected some indirect supporting evidences:
> - Snowflake Arctic independently verified our approach, by using a 3-stage curriculum inspired by us (start with web text, then move to domain-specific (coding) ones): https://www.snowflake.com/blog/arctic-open-efficient-foundation-language-models-snowflake/.
> - Olmo-7B and another of our 7B models, both trained under similar setups, show significantly lower MMLU compared to Crystal at a similar token scale.
>
> Research on pre-training is challenging, especially for detailed ablation studies. This is why we open-source the creation process of models, to empower the community to study this collectively.
>
> **Additional Comparison**
> - *GPT-3.5/4*: These models update constantly, and their model details and conditions (e.g., model size, instruction data used) are unknown. But we agree that adding them as references would provide a reference point.
> - *Mixtral 8x7B*: We compared Crystal with models of similar settings. Mixtral, a mixture-of-expert model, has a significantly larger effective parameter size, and an unknown token budget. We will include a discussion addressing the comparison with such models in our revised version.
>
> **Novelty of the method**
> - Pretraining research often focuses on scaling up straightforward ideas, we are happy to contribute to a simple yet effective method.
> - Due to resource limitations, we only conducted a two-phase experiment and found that a well-designed data curriculum benefits pretraining. We hope this can inspire future research into multi-phase or continuous pretraining.
> - Additionally, we released a full training trace, which, to the best of our knowledge, provides a unique resource for the community to build upon variations of our method.
>
> **Additional metrics**\
> We have taken your suggestion into account and have begun setting up evaluations for more coding tasks.

---

> > ### Comment · Reviewer_zzp6 · 2024-05-31
> > **Thank you for your response**
> >
> > Thank you for your response and clarifications. I decide to keep my original review score.

---

### Official Review · Reviewer_mxyS · 2024-05-12

**Rating:** 6
**Confidence:** 5
**Ethics Flag:** 1

**Summary:**

Authors propose a pre-training strategy that transitions from a phase of learning from natural language heavy to code heavy dataset. This has been claimed to achieve the most optimal balance between performance on natural language tasks and coding tasks for a smaller training budget than comparable work (in terms of model size) like Llama2 and CodeLlama.

**Questions To Authors:**

- The CG-1 architecture is not described in sufficient detail. Besides LayerNorm what architectural changes are required for training on this non-GPU hardware?
- What is the training duration? (similar to how GPU hours are being reported in other studies)
- In Fig 3, can the authors explain the occurrence of spikes in the training loss during stage 1?
- Why is the qk dot product scaling different compared to the commonly used $\sqrt{d}$ formula? (Table 4)
- What does rotary percentage mean in Table 4?
- What is the source of the 100B tokens in the phase adaptation stage? What is the motivation in including this stage?
- Creating a benchmark like WebMC is an interesting contribution. Can the authors provide qualitative analysis to describe what the tasks here look like?
- In 5.2 authors mention the adaptation phase to be hurting NL tasks. Could the reason be that the fraction of Slim Pajama is too little in this stage?

Suggestions:
- Pre-trained models like X-Gen, Phi series could be added to the related work section. XGen (by Nijkamp et al https://arxiv.org/abs/2309.03450) specifically discusses the approach of having different stages each with different sampling proportions.
- Besides k-shot, authors could also try CoT prompting w/ the Crystal models to push performance on the NL and code benchmarks, particularly MATH and GSM8k.
- A summary of model architecture details could be a part of the main paper (currently in Table 5 and Appendix B), similarly Table 7 should also be a part of the main paper. There is quite some whitespace (e.g. Fig 5 (could use shared legends), Fig 6, Section 6) that could be removed to achieve this compression.

Minor typos/linguistic issues:
- Abstract: Futhermore, there lack of --> there is a lack of?
- Section 3 Phase 2: we intent to --> we intend to?
- Section 5.2 Para 2: adaptation gains are even out

**Reasons To Accept:**

This work provide a closer look at the details of the pre-training stages of language models, which has been undisclosed in the case of most openly released models in recent times. Particularly, the stages of pre-training and their impact on downstream performance through empirical studies haven't been studied in this manner to the best of my knowledge. This can benefit the community's understanding in efficiently training models (with as small pre-training costs as possible) that are performance-wise optimal in both code and natural language domains. For instance, they highlight the variation in the loss scales across the two phases as code data is more predictable leading to a lower loss-per-token. The final model they obtain after the different training stages is indeed competitive in both domains when compared to specialised models like Llama 2 and CodeLlama that have been trained on a similar scale of data.

**Reasons To Reject:**

Authors don't discuss the motivation of needing a single model to perform well at tasks from both natural language and code domain. One possible motivation could be that Code LLMs that can follow instructions are suited for a wide variety of agentic tasks as shown in Hong et al. However, one could argue if this capability can be achieved simply by instruction fine-tuning stage on top of a code pre-trained LM.

While the impact of controlling the ratio of code to natural language data during pre-training has been studied, there is no discussion on the quality of the dataset used. Recent work on phi-2 (https://www.microsoft.com/en-us/research/blog/phi-2-the-surprising-power-of-small-language-models/) has shown performance superior (MMLU, HumanEval, MBPP) to Crystal when trained on a similar size of data ~1.4T tokens, and using a smaller model in terms of parameter count (2.7B). Comparing this work to phi-3 or Llama 3 wouldn't be fair given the timeline of events.

Hong et al MetaGPT: Meta Programming for A Multi-Agent Collaborative Framework

---

> ### Author Rebuttal · Authors · 2024-05-31
>
> Thank you for your thoughtful reviews!
>
> **Motivation**
> We’ve addressed the need for a single model in the introduction. While code LLMs can improve their natural language (NL) skills through finetuning, their NL proficiency may suffer in broader domains. Our goal is to develop a model that generally excels in both, proven by the performance of our models.
> Moreover, there is a hypothesis that code and NL can enhance each other. Though mentioned in prior works, it hasn’t been documented in pretraining literature. This research shows that the interplay between NL and code can indeed boost performance, even using less tokens.
> Finally, the ability to mix different types of data enables further scaling the total token size.
>
> **Data quality and Phi**
> The focus of this work is multi-stage and the interplay between NL and code. While we acknowledge that data quality is essential, it is orthogonal to our research goal. Our training strategy can be effectively combined with higher-quality data.
>
> The highlight of Phi-2 is its synthetic data, whereas we use publicly available corpora (SlimPajama, Starcoder) with validated quality. This saves time on massaging prompts, and compute on querying powerful LLMs. Moreover, the open source data makes our research more accessible and verifiable.
>
> **Questions**
> - RMSNorm is to resolve a computation bottleneck unique to GPU, which doesn’t exist on CS-2, allowing us to use Layernorm
> - 37 days on 16 CS-2 nodes
> - Loss spikes are commonly observed, while the explanation remain open. Spikes in Crystal pretraining is not frequent compared to other efforts (e.g. Llama). When a major loss spike is observed, it can be handled by skipping a specific data batch
> - RoPE applied to first 25% of embedding dimensions, inspired by GPT-NeoX
> - Based on our experiments, q*k dot product works better with muP
> - Prior work (e.g. CodeLlama) showed that a stage targeting a language (e.g., python) can boost performance. Our goal is to improve Python/Web programming. Data is sampled from Starcoder V1
> - WebMC has 3 task types:
>   - generation: generate a websites with specifics (e.g., has a map)
>   - editing: change font-family/font-color
>   - understanding: ask about element details, such as what’s in the nav bar
> - Your hypothesis is inline with ours that the last stage’s data mix might be the culprit, which implies continuous training can be sensitive to the model’s data curriculum
>
> We will add more details in the revision to address the questions.

---

> > ### Comment · Reviewer_mxyS · 2024-06-06
> > **Response to rebuttal**
> >
> > I thank the authors for the clarifications provided, and am revising my score accordingly.

---

### Official Review · Reviewer_oMAi · 2024-05-12

**Rating:** 6
**Confidence:** 4
**Ethics Flag:** 1

**Summary:**

The authors introduced a pretraining strategy that focuses on training LLMs with both text and code data. The resulting family of models, CRYSTAL, is stronger in both natural language and coding performance, compared to strong baselines like LLama2 and Code Llama. The authors are also committed to releasing open-sourced model checkpoints, increasing the replicability and transparency of LLM training.

**Questions To Authors:**

- Typo (Introduction): “pertaining”
- A useful ablation the authors can try is to use the data in phase 1 or phase 2 but train for the same number of training steps as phase 1 + phase 2 training combined. This would isolate the effect of training steps and demonstrate better how the data mixing contributes to the performance.

**Reasons To Accept:**

- Despite pretrained on a smaller set of tokens (1.4 trillion), CRYSTAL seems to possess strong/comparable performance against other models pretrained on a larger dataset (2 trillion tokens).
- The evaluation is done comprehensively on many language tasks (reasoning, QA, general knowledge, etc.) and coding tasks (HumanEval, MBPP, WebMC). Interesting ablation results (e.g. data efficiency, training curves) are analyzed quite well, bringing useful insights to LLM training.
- The authors’ commitment to open-source the models is commendable as it will help to push innovation and collaboration within the research community.

**Reasons To Reject:**

- The use of a mixture of datasets of code and text over multiple stages of training is not so new. For instance, CodeT5+ (https://arxiv.org/abs/2305.07922) was also trained on both code and text data over two stages of pretraining. While the model was not evaluated on conventional NLP tasks, it performed quite well over multimodal test setups such as code-to-text and text-to-code benchmarks. Can the authors clarify what is the main novelty compared to CodeT5+? Have you tried to benchmark CRYSTAL in more diverse code-text tasks beyond HumanEval and MBPP?
- Pretraining models from scratch is expensive. I wonder if the current pretraining strategy can be applied on top of any pre-trained models. If subsequently similar insights could be observed, I think the current method would be a lot more impactful.

---

> ### Author Rebuttal · Authors · 2024-05-31
>
> Thank you for the thoughtful review and helpful feedback! We respond to individual comments below.
>
> **Main Novelty Compared to CodeT5+**\
> CodeT5+ starts with pre-trained LLMs, similar to CodeLlama. As we argued in the paper, such model development strategies can hurt the original language capabilities injected in the pre-trained LLMs, resulting in the final code LLM not achieving a perfect balance between language and code capabilities. This, in fact, is the key advantage of CrystalCoder.
>
> **Benchmark Crystal with more diverse coding tasks beyond HumanEval and MBPP**\
> Thanks a lot for the great suggestion! We are actively working on evaluating Crystal on APPS and DS-1000 as suggested. Due to the time constraint, we don’t have any number to share yet. But we will post the results as soon as they are ready.
>
> **Current strategy applied on top of other pretrained models** \
> Whether our proposed strategy is compatible with pre-trained models remains an open question. Knowing the data composition of pre-trained LLMs is crucial to avoid issues such as introducing duplicated text to a model or using a wrong data weight. The lack of transparency from existing LLMs hinders this research. Additionally, conducting ablation studies becomes challenging without knowledge of their specific settings, as many open-weight LLMs only release their final model weights. Our fully open-sourced Crystal model provides a foundation for the community to conduct these experiments.
>
> **Question: ablation study with same number of tokens**
> - We agree with the reviewers that it would be great to include explicit ablation studies. However, pre-training is expensive, which is unfortunately hard to afford. We have started the ablation study as suggested by the reviewer but due to resource constraints, it is not finished.
> - Luckily, our approach has been verified by another team independently and has demonstrated strong performance. For example, Snowflake Arctic used a three-stage curriculum learning strategy inspired by our approach (starting with a focus on generic skills, then moving to more domain-specific knowledge and skills): https://www.snowflake.com/blog/arctic-open-efficient-foundation-language-models-snowflake/.
> - Olmo-7B and another of our 7B models, both trained under similar setups, show significantly lower MMLU compared to Crystal at a similar token scale. While not a direct comparison, these results further support the effectiveness of our strategy.

---

> > ### Comment · Reviewer_oMAi · 2024-06-04
> >
> > Thanks to the authors for replying to my questions. I will keep my original review score. I look forward to the additional results and the full release of the models, both would significantly help the research community with useful insights.

---

### Decision · Program_Chairs · 2024-07-10

**Decision:**

Accept

**Comment:**

This paper presents a 2-phase pretraining process with the goal of training a single model that excels at both natural language and code. Basically, phase 1 contains mostly text with a small proportion of code, while phase 2 includes more code. They demonstrate that their Crystal model achieves competitive performance on both text and code tasks compared to other open-weight models of a similar size.

This work presents insights into the effect of different pretraining stages, and it would be a valuable contribution if the authors can fully open-source their model checkpoints and code. I recommend the authors to incorporate reviewers' feedback into their camera-ready version, especially with new experimental results they promised to add.